# Erectile dysfunction after COVID-19 recovery: A follow-up study

**Kawintharat Harirugsakul[1], Sorawit Wainipitapong[2], Jeerath Phannajit[3], Leilani Paitoonpong[4], Kavirach Tantiwongse[1] ***

1 Division of Urology, Department of Surgery, Faculty of Medicine, Chulalongkorn University and King Chulalongkorn Memorial Hospital, The Thai Red Cross Society, Bangkok, Thailand, 2 Department of Psychiatry and Center of Excellence in Transgender Health, Faculty of Medicine, Chulalongkorn University and King Chulalongkorn Memorial Hospital, The Thai Red Cross Society, Bangkok, Thailand, 3 Division of Clinical Epidemiology and Division of Nephrology, Department of Medicine, Faculty of Medicine, Chulalongkorn University and King Chulalongkorn Memorial Hospital, The Thai Red Cross Society, Bangkok, Thailand, 4 Division of Infectious Diseases, Department of Medicine, Faculty of Medicine, Chulalongkorn University and King Chulalongkorn Memorial Hospital, The Thai Red Cross Society, Bangkok, Thailand

* Kavirach@chula.md

## Abstract

### Objectives

Several studies confirm multiple complications after COVID-19 infection, including men's sexual health, which is caused by both physical and psychological factors. However, studies focusing on long-term effects among recovered patients are still lacking. Therefore, we aimed to investigate the erectile function at three months after COVID-19 recovery along with its predicting factors.

### Methods

We enrolled all COVID-19 male patients, who were hospitalized from May to July 2021, and declared to be sexually active within the previous two weeks. Demographic data, mental health status, and erectile function were collected at baseline and prospectively recollected three months after hospital discharge. To determine changes between baseline and the follow-up, a generalized linear mixed effect model (GLMM) was used. Also, logistic regression analysis was used to identify the associating factors of erectile dysfunction (ED) at three months.

### Results

One hundred fifty-three men with COVID-19 participated. Using GLMM, ED prevalence at three months after recovery was 50.3%, which was significantly lower compared with ED prevalence at baseline (64.7%, P = 0.002). Declination of prevalence of major depression and anxiety disorder was found, but only major depression reached statistical significance (major depression 13.7% vs. 1.4%, P < 0.001, anxiety disorder 5.2% vs. 2.8% P = 0.22). Logistic regression, adjusted for BMI, medical comorbidities, and self-reported normal morning erection, showed a significant association between ED at three months and age above

**Funding:** The authors received no specific funding for this work.

**Competing interests:** The authors have declared that no competing interests exist.

40 years and diagnosis of major depression with adjusted OR of 2.65, 95% CI 1.17–6.01, P = 0.02 and 8.93, 95% CI 2.28–34.9, P = 0.002, respectively.

## Conclusion

Our study showed a high ED prevalence during the third month of recovery from COVID-19. The predicting factors of persistent ED were age over 40 years and diagnosis of major depression during acute infection.

## Introduction

Since December 2019, severe acute respiratory syndrome coronavirus-2 (SARS-CoV-2) has been spreading worldwide. Because of the virus' highly contagious rate and mutation ability, nowadays, more than 440 million people are infected in 150 countries [1]. Essentially, sequelae of COVID-19 were reported in the multiorgan system throughout the body [2], and several studies found that the virus negatively affected male reproductive health and sexual function [3, 4]. For erectile dysfunction (ED), the etiology of its impacts is believed to be contributed by multi-dimensional causes, and the outcome after recovery remains mysterious regarding limited data about long-term effects, biologically in particular, of COVID-19 infection [5].

Pathophysiology of ED includes biological and psychological etiology [6]. SARS-CoV-2 can affect erectile function via various methods, and it may also target the genitourinary system by entering host cells through angiotensin-converting enzyme 2 receptor of vascular endothelial cells. Accordingly, the hyperinflammation stage from the secretion of inflammatory cytokines (TNF-$\alpha$, IL-6, and IL-1), a thrombo-embolic phenomenon and effect on other organ systems, especially endocrine, may lead to ED [7]. A study reported the presence of SARS-CoV-2 in penile vascular endothelial cell, crucial to penile erection, of post-infected COVID-19 patients with severe ED. The study also reported a decreased nitric oxide synthase expression in corpus cavernosum, which can be a consequence of endothelial dysfunction [8].

Besides physical causes mentioned above, psychosocial sequelae of COVID-19, including socioeconomic and mental problems, also alter men's sexual health [5, 9]. COVID-19 pandemic leads to economic hardship, including job loss and financial burden, which are critical psychosocial stressors. They can co-exist with multiple psychiatric morbidities such as depression, anxiety, stress, suicidal ideation, and sleep problems [10–12]. Moreover, the reduction of sexual intercourse frequency, which might represent the sexual impacts of COVID-19, is reported during this pandemic [12]. Some online surveillances indicated a deteriorated erectile function and sexual dissatisfaction globally [14], and a higher prevalence of ED was observed compared with the matched population [15].

Even though we currently understand more about COVID-19, the complication after recovery should receive much more academic attention. Long COVID or post-COVID syndrome [16] is still poorly understood but believed to affect the quality of life regarding individual's risk factors. Clinicians worldwide recognized this persistent discomfort and disability and, consequently, some guidelines, such as National Institute for Health and Care Excellence (NICE) and CDC (Center for Disease Control), toward Long COVID are now established [17, 18].

Long COVID might be termed 'symptomatic syndrome beyond three weeks after COVID-19 infection' [19]. The largest descriptive study reported that 37.7% among 508,707 recovered patients have at least one symptom at the twelfth week. Risk factors of long COVID include biologically female gender, high body mass index (BMI), smoking, and having lower incomes;

meanwhile, Asian ethnicity is a protective factor [20]. A systematic review of 57 studies showed that the rate of post-acute sequelae of COVID-19 at six months is approximately 54%. Long COVID patients show systemic sequelae, and abnormal chest imaging remained the most frequent pulmonary sequelae (median [IQR], 62.2% [45.8%-76.5%]). Neuropsychiatric consequences are also common but have a lower prevalence: difficulty concentrating (median [IQR], 23.8% [20.4%-25.9%]) and generalized anxiety disorder (median [IQR], 29.6% [14.0%-44.0%]) [21].

Sexual long COVID syndrome, referred to as persisting sexual dysfunction after COVID-19 infection [5], is a concerning problem that may affect the quality of life. Nonetheless, it is being endured in silence due to clinicians' disregard, especially in the Asian context where sexual problems are underrecognized and avoided [22].

Studies about this syndrome are still limited. One research indicated the prevalence of ED at 6–9 months after recovery [23]. However, further post-COVID ED studies are also necessary regarding differences in sociocultural contexts or periods after recovery. Also, this might be beneficial in predicting other long-term complications since ED could reflect other organ functions, a cardiovascular system in particular, as the "tip of the iceberg" [24]. An earlier ED presentation, such as three months after COVID-19 recovery could be advantageous for both clinicians and patients for raising awareness of further complications afterward. Thus, our study aimed to examine the three-month changes in erectile function and its associated factors among patients who recover from COVID-19 infection.

## Materials and methods

### Study design and participants

This is the observational cohort study of COVID-19 male patients hospitalized at King Chulalongkorn Memorial Hospital, the Thai Red cross society, one of the largest university hospitals in Bangkok, Thailand. Between May and July 2021, we enrolled all male patients aged between 18–70 years old, tested COVID-19 positive for a reverse transcriptase-polymerase chain reaction assay using nasopharyngeal swab specimen, and declared being sexually active within the previous two weeks. Patients with severe medical and mental conditions were excluded.

The study was registered with the Thai Clinical Trials Registry, with case number TCTR20210617008, and was approved by Chulalongkorn University's Institutional Review Board (COA No. 659/2021). The sample size of seventy-eight participants at a minimum was calculated using the previous study's ED prevalence [15].

### Data collection

**Study design.** From May 2021, patients according to the inclusion and exclusion criteria were invited to participate. Inform consent was obtained prior to the assessment. The assessment was conducted online or through a phone interview with illiterate patients to prevent viral transmission. All participants were reassessed three months after recovery by the same questionnaires. Then, the study excluded individuals reporting being sexually inactive.

**Demographic data.** At baseline, recorded demographic included age, BMI, underlying diseases, education and marital status, and history of alcohol and nicotine use. Details about COVID-19 vaccination and treatment during hospitalization were extracted.

During and three months after the COVID-19 infection, all participants' erectile function was assessed by the Thai version of the International Index of Erectile Function 5 (IIEF-5) [25]. Mental health status was screened by the Thai Patient Health Questionnaire 9 (PHQ-9) for major depression [26] and the Generalized Anxiety Disorder Scale (GAD-7) for anxiety [27].

In addition, individuals' report of penile morning erection was accumulated in our study.

**Assessment of erectile function.** Thai IIEF-5 is a self-rated measurement containing five questions focusing on erectile function and sexual intercourse satisfaction, validated in the Thai population. It is a standard assessment tool for ED assessment and diagnosis. Its scores negatively correlated with ED severity and could be classified into five levels: severe [5–7], moderate [8–11], mild to moderate [12–16], mild [17–21], and no ED [22–25].

**Assessment of mental health status.** Thai PHQ-9 was used to evaluate depressive symptoms, including depressed mood, loss of interest and energy, sleep and appetite problems, feelings of worthlessness, trouble concentrating, psychomotor abnormalities, and thoughts of death or self-injury. A PHQ-9 score of $\geq 9$ is considered positive for major depression, with sensitivity and specificity of 0.84 and 0.77, respectively. The test was validated in the Thai population and showed good psychometric properties [26].

The GAD-7 scale is an assessment for generalized anxiety disorder and anxiety severity. The measured symptoms included nervousness, uncontrolled and excessive worrying, trouble relaxing, restlessness, irritability, and fear that something awful might happen. A GAD-7 score of $\geq 10$ represents a cut point for identifying generalized anxiety disorder with sensitivity and specificity of 0.89 and 0.82, respectively [27].

## Statistical analysis

Categorical variables were presented as counts with percentages; meanwhile, continuous variables were reported as mean with standard deviation (SD) for normally distributed data or median with interquartile range (IQR) for non-normally distributed data. Shapiro-Wilk test was used to test for normality. All non-normally distributed data were log-transformed before comparison. To determine the difference in clinical characteristics between the baseline and the follow-up cohort, in which some participants were lost, the independent t-test and chi-square test were used to compare continuous and categorical variables, respectively. Temporal changes of erectile dysfunction status and psychological parameters after the 3-month recovery from COVID-19 infection were tested using a linear mixed effect model (LMM) for continuous data including IIEF-5, GAD-7, and PHQ-9 scores using visit (baseline vs. 3-month) as a covariate and participant ID as fixed intercept. Binary variables, including ED status, presence of normal morning erection, diagnosis of anxiety, and major depression were tested using a generalized linear mixed effect model (GLMM) with the same covariate and fixed intercept as the LMM. To determine the association between the 3-month post-recovery ED and relevant factors, bivariate and multivariable logistic regression were used. Factors with a P value below 0.2 from the bivariate analysis and those with theoretically ED associated were recruited to the multivariable logistic regression. Adjusted odds ratios (OR) were reported with 95% confidence intervals (95%CI). After the 3-month follow-up, participants were classified into four groups based on ED statuses: no ED, transient ED (ED at baseline and without ED at the follow-up), Persistent ED (ED at baseline and the follow-up), and later onset ED (ED at the follow-up without ED at baseline). One-way analysis of variance (ANOVA) and chi-square test were used to compare continuous and categorical clinical characteristics between these four groups, respectively. A P value of $< 0.05$ was considered statistically significant. All P values were unadjusted for multiplicity since only exploratory analyses were planned. All analyses were performed using STATA-IC Version 16.1.

## Results

From May to July 2021, 153 from 654 hospitalized COVID-19 male patients had reported being sexually active and eligible for the study. During the admission, the first assessment was

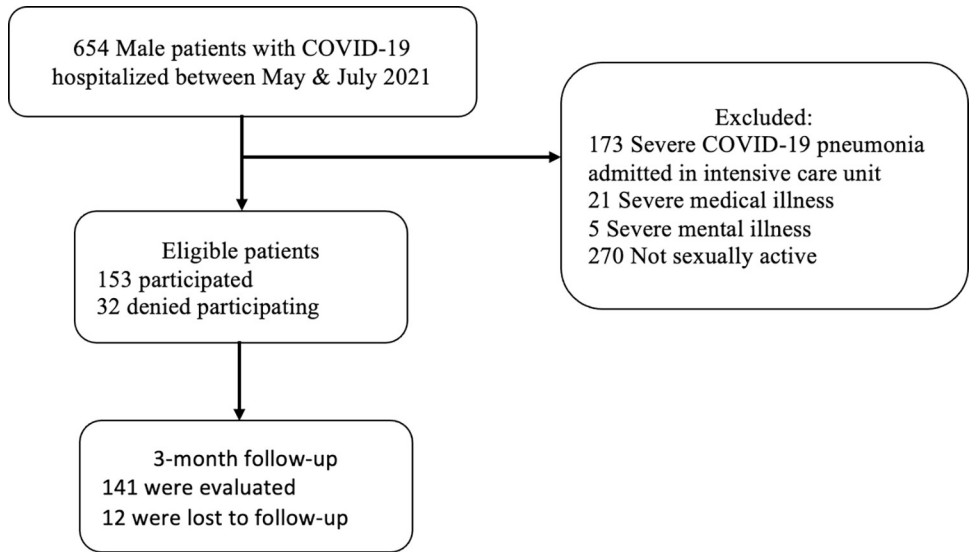

**Fig 1. Study protocol.**

done, and their COVID-19 treatment history until their discharge was collected. Three months after the COVID-19 infection, a completed second evaluation was obtained from 141 participants (Fig 1).

Table 1 displays demographic data at each point of the assessment. Mean age (40.8 and 41.0 years), BMI (25.6 and 25.7 kg/m$^2$), and other characteristics showed no significant differences between the first and second evaluations. Most were married or in a relationship as a couple, employed, and had educational attainment lower than a bachelor's degree.

Erectile function, self-reported normal morning erection and mental health status, including anxiety and depression, were evaluated at baseline. After reassessment at three months following COVID-19 recovery, we found statistically significant differences in all dimensions except prevalence of anxiety disorder and self-reported normal morning erection (Table 2).

To determine factors associated with ED at three months, all participants whose IIEF-5 results were positive for ED underwent a logistic regression (Table 3). Variables with significant association (P < 0.2) from the bivariate model were additionally analyzed in the multivariable logistic regression. We also considered the age and medical comorbidities, which were notable ED risk factors. Age over 40 years old (adjusted OR 2.65, 95% CI 1.17–6.01, P = 0.02) and major depression during infection (adjusted OR 8.93, 95% CI 2.28–34.9, P = 0.002) were significantly associated with ED at three months; meanwhile, overweight (P = 0.05) and presence of hypertension, diabetes mellitus, or hypercholesterolemia (P = 0.06) were nearly significant.

As ED is a dynamic disorder, changing over time [28] and there are still no gold standard categorizing ED related with COVID-19. Thus, we categorized all participants into four groups regarding their ED course and onset; no ED, transient ED, persistent ED, and later onset ED. Persistent ED was the most prevalent (42.6%) and related to major depression during infection. A statistically significant difference in self-reported normal morning erection after recovery was also found (P = 0.01). Table 4 shows the demographic data and associated factors among each group.

## Discussion

As COVID-19 continues to spread globally, it is predicted that it will infect more than half of the world population [17]. Clinicians should prepare to overcome complications after

**Table 1. Demographic data at baseline and three months after COVID-19 recovery.**

| Variables | Baseline[†] (N = 153) | 3 months[†] (N = 141) | P value[‡] |
|---|---|---|---|
| Age (mean ± SD) | 40.8 ± 10.9 | 41.0 ± 10.5 | 0.81 |
| Body mass index (mean ± SD) | 25.6 ± 4.4 | 25.7 ± 4.4 | 0.88 |
| Marital status | | | 0.98 |
| • Single | 25 (16.3%) | 22 (15.6%) | |
| • Married or couple | 125 (81.7%) | 116 (82.3%) | |
| • Separated or divorced | 3 (2.0%) | 3 (2.1%) | |
| Education | | | 0.88 |
| • Lower than bachelor's degree | 129 (84.3%) | 118 (83.7%) | |
| • Bachelor's degree or upper | 24 (15.7%) | 23 (16.3%) | |
| Medical comorbidities | 29 (19%) | 28 (19.9%) | 0.84 |
| • Diabetes Mellitus | 13 (8.5%) | 13 (9.2%) | 0.83 |
| • Hypertension | 16 (10.5%) | 15 (10.6%) | 0.96 |
| • Hypercholesterolemia | 11 (7.2%) | 11 (7.8%) | 0.84 |
| Substance use | | | |
| Active alcohol drinking | 26 (17.0%) | 24 (17.0%) | 0.99 |
| Active smoking | 57 (37.3%) | 51 (36.2%) | 0.85 |
| COVID-19 vaccination | 50 (32.7%) | 48 (34%) | 0.80 |
| Severity of COVID-19 | | | 0.94 |
| Pneumonia | 63 (41.2%) | 60 (42.6%) | |
| Pharyngitis | 79 (51.6%) | 70 (49.6%) | |
| Asymptomatic | 11 (7.2%) | 11 (7.8%) | |
| COVID-19 treatment | | | |
| Favipiravir | 70 (45.8%) | 65 (46.1%) | 0.95 |
| Corticosteroids | 36 (23.5%) | 36 (25.5%) | 0.69 |
| Remdesivir | 10 (6.5%) | 10 (7.1%) | 0.85 |
| Tocilizumab | 5 (3.3%) | 5 (3.5%) | 0.90 |
| Oxygen supplement | 20 (13.1%) | 20 (14.2%) | 0.78 |

[†] Data were shown in counts (%) for categorical variables and mean ± standard deviation (SD) for continuous variables.

[‡] P value for unpaired difference between two groups using chi-square test for categorical variables and independent t-test for continuous variables in order to determine the differences between baseline and the follow-up cohorts.

COVID-19 infection or long COVID, which is considered common (37.7–54.0%) in recovered patients [20, 21].

Despite several etiologies or pathogenesis of long COVID, multiple studies hypothesized that long COVID might be due to several factors. The factors include incomplete viral eradication, prolonged inflammatory response resulting from poor immune system and residual viral remnant, or direct viral infiltration into organ system and cytokine network dysregulation [29–31] Additionally, ED, as a consequence of long COVID, is also complicated by both aforementioned biological and notorious psychological factors including socioeconomic problems, social isolation, or traumatic events.

Notably, impaired erectile function was reported to be a part of sexual long COVID (SLC) [5]. We found that the prevalence of ED at the third month compared with during COVID-19 infection, was significantly lower (50.3% vs. 64.7%, P = 0.002), which implied the erectile function improvement. However, the number was still higher than ED in the general Thai population (37.5–42.2%) [32, 33]. Association between lower prevalence and a longer period after recovery was also reported in one study from China with ED prevalence of 44.8% and 30%, at six and nine months after recovery of COVID-19, respectively [23].

**Table 2. Temporal changes in erectile function and mental health status at baseline and three months after COVID-19 recovery (N = 141).**

| Variables | Baseline[†] (N = 153) | 3 months[†] (N = 141) | P value[‡] |
|---|---|---|---|
| **Erectile dysfunction** | 99 (64.7%) | 71 (50.3%) | 0.002[(b)] |
| **IIEF-5 score** | 21 [18–23] | 20 [18–22] | 0.02[(a)] |
| **Anxiety disorder (GAD-7 ≥ 10)** | 8 (5.2%) | 4 (2.8%) | 0.22[(b)] |
| **GAD-7 score** | 2 [1–5] | 1 [0–4] | 0.03[(a)] |
| **Major depression (PHQ-9 ≥ 9)** | 21 (13.7%) | 2 (1.4%) | < 0.001[(b)] |
| **PHQ-9 score** | 3 [1–7] | 0 [0] | 0.002[(a)] |
| **Normal morning erection** | 129 (84.3%) | 125 (88.7%) | 0.10[(b)] |

Abbreviations: IIEF-5 = International Index of Erectile Function 5; GAD-7 = General Anxiety Disorder Scale PHQ-9 = Patient Health Questionnaire 9.

[†] Data were shown in counts (%) for categorical variables and median [interquartile range] for continuous variables.

[‡] P value for difference between two visits using (a) linear mixed effect model for log-transformed continuous variables and (b) generalized linear mixed effect model for binary variables.

We speculate that improvement of ED could be explained by gradual recuperation of physical competency. Apart from ED, other symptoms during COVID-19 infection, including mental health, anosmia and ageusia, were also significantly improved after three months of infection [34–36].

Moreover, mentality during infection might be accountable for the higher prevalence of ED. Physical stress that became intact after recovery and psychosocial difficulties that tended to be relieved at three months also led to better mental status and resulted in better erectile function. Bidirectionally, improvement in sexual function could enrich sexual health and, consequently, promote psychological well-being [9]. This confirms the complexity between ED and mental health which requires further bio-psycho-social investigations. In addition, studies on SLC are still limited and its risk factors are questionable. Thus, future SLC research would be immensely beneficial [37].

**Table 3. Associated factors of erectile dysfunction after three months after recovery from COVID-19 infection using bivariate and multivariable logistic regression model.**

| Variables | Bivariate model | | Multivariable model | |
|---|---|---|---|---|
| | Crude OR (95%CI) | P value | Adjusted OR (95% CI) | P value |
| **Age (years)** | 1.01 (0.98–1.04) | 0.625 | | |
| **Age over 40 years** | 1.54 (0.79–2.99) | 0.208 | 2.65 (1.17–6.01) | 0.02 * |
| Body mass index (kg/m²) | 0.96 (0.89–1.04) | 0.318 | | |
| **Overweight (Body mass index ≥ 23)** | 0.61 (0.28–1.30) | 0.199 | 0.43 (0.18–1.00) | 0.05 |
| **Married or couple** | 0.92 (0.39–2.19) | 0.856 | | |
| **Active smoking** | 1.18 (0.59–2.34) | 0.644 | | |
| **Active alcohol drinking** | 0.98 (0.41–2.37) | 0.970 | | |
| **Medical comorbidities (HT, DM, HCL)** | 0.82 (0.36–1.88) | 0.643 | 0.37 (0.13–1.06) | 0.06 |
| **COVID-19 vaccination** | 1.43 (0.71–2.89) | 0.315 | | |
| **COVID-19 pneumonia** | 1.23 (0.63–2.40) | 0.543 | | |
| **Anxiety disorder (baseline)** | 3.14 (0.61–16.12) | 0.171 | 2.62 (0.34–20.39) | 0.36 |
| Major depression (baseline) | 7.58 (2.12–27.12) | 0.002 | 8.93 (2.28–34.9) | 0.002 * |
| Normal morning erection (baseline) | 0.55 (0.22–1.36) | 0.195 | 0.49 (0.19–1.31) | 0.15 |

Abbreviations: HT–Hypertension, DM–Diabetes mellitus, HCL–Hypercholesterolemia.

* P value < 0.05

**Table 4. Demographic data and associated factors among four groups of erectile dysfunction.**

| Variables[†] | No ED (N = 37) | Transient ED (N = 33) | Persistent ED (N = 60) | Later onset ED (N = 11) | P value[‡] |
|---|---|---|---|---|---|
| **Age (years)** | 41.1 ± 11.2 | 40.1 ± 9.8 | 42.7 ± 10.4 | 34.6 ± 9.7 | 0.11 |
| **Body mass index (kg/m$^2$)** | 25.6 ± 4.0 | 26.5 ± 4.4 | 25.7 ± 4.6 | 23.4 ± 4.5 | 0.24 |
| **Medical comorbidities** | 8 (21.6%) | 7 (21.2%) | 12 (20.0%) | 1 (9.1%) | 0.82 |
| • Diabetes mellitus | 3 (8.1%) | 3 (9.1%) | 6 (10.0%) | 1 (9.1%) | 0.99 |
| • Hypertension | 6 (16.2%) | 3 (9.1%) | 5 (8.3%) | 1 (9.1%) | 0.65 |
| • Hypercholesterolemia | 2 (5.4%) | 2 (6.1%) | 6 (10.0%) | 1 (9.1%) | 0.83 |
| **Substance use** | | | | | |
| Active alcohol drinking | 8 (21.6%) | 4 (12.1%) | 10 (16.7%) | 2 (18.2%) | 0.77 |
| Active smoking | 11 (29.7%) | 13 (39.4%) | 22 (36.7%) | 5 (45.5%) | 0.75 |
| **COVID-19 vaccination** | 12 (32.4%) | 9 (27.3) | 23 (38.3) | 4 (36.4) | 0.74 |
| **Anxiety disorder (baseline)** | 0 (0%) | 2 (6.1%) | 6 (10.0%) | 0 (0%) | 0.17 |
| **Anxiety disorder (3 months)** | 0 (0%) | 0 (0%) | 4 (6.7%) | 0 (0%) | 0.14 |
| **Major depression (baseline)** | 0 (0%) | 3 (9.1%) | 17 (28.3%) | 1 (9.1%) | 0.001 [*] |
| **Major depression (3 months)** | 0 (0%) | 0 (0%) | 2 (3.3%) | 0 (0%) | 0.43 |
| **Normal morning erection (baseline)** | 30 (81.1%) | 31 (93.9%) | 47 (78.3%) | 9 (81.8%) | 0.28 |
| **Normal morning erection (3 months)** | 36 (97.3%) | 31 (93.9%) | 51 (85.0%) | 7 (63.6%) | 0.010 [*] |

[†] Data were shown in counts (%) for categorical variables and mean ± standard deviation for continuous variables.

[‡] P value for difference between groups using chi-square test for categorical variables and one-way analysis of variance for continuous variables.

[*] P value < 0.05

Adjusted for BMI, medical comorbidities, anxiety severity, and normal morning erection, multivariable logistic regression identified age over 40 years old (adjusted OR 2.65, 95% CI 1.17–6.01, P = 0.02) and having major depression during infection (adjusted OR 8.93, 95% CI 2.28–34.9, P = 0.002) as predicting factors of ED at three months. Both older age and depression are established risk factors for ED [38, 39] and our result emphasized the role of a bio-psychological issue in the pathogenesis of ED. Interestingly, only mental conditions without aging were found to be associated with ED during infection [40], but patients over 40 years of age were also at risk for ED at three months. Elderly COVID-19 patients should be screened for SLC, and long-term follow-up is still necessary.

ED course found in our participants was both persistent and self-remitted. This was similar to the prognosis of other long COVID complications, which were wax and wane or uncertain [41]. Improvement of COVID-19 symptoms, especially anosmia and ageusia, which contributed to ED etiologies could explain a recovery in patients whose ED was transitory [42].

To our knowledge, our study was the first ED cohort study on COVID-19 patients for three months after recovery. Our sample size was considerable, and all participants were confirmed COVID-19 diagnosis by a gold standard method. Both biological and psychosocial aspects were measured, and our dropout rate was low. However, this study is not without limitations. Firstly, because of a lack of comparison group and pre-existed erectile function status, the assumption of whether COVID-19 was the cause of ED could not be concluded and the confounding bias from social situations such as health policy during pandemic and change in sexual habit might affect the erectile function. Nevertheless, the improvement of ED three months after COVID-19 recovery could be assumed from our study except in older and major depression patients who needed further monitoring. Secondly, limited generalizability should be declared, considering all participants were hospitalized and those with severe symptoms were excluded. The questionnaire used in our study was self-rated and affected by a recall bias.

Treatment of persistent ED should be further studied to help clinicians and patients globally, as the number of COVID-19 patients, both recovered and infected, is still increasing.

## Conclusion

Although long COVID has been widely studied, only a few studies have focused on erectile function as its complication. Our study showed that, even though the erectile function was significantly improved after three months of COVID-19 infection, the prevalence of ED was still high. In addition, male patients older than 40 years or having major depression during COVID-19 were at risk to be screened positive for ED at three months. Future studies focusing on ED treatment, especially in persistent ED, would be helpful for both clinicians and patients in the time after pandemic cessation.

## Acknowledgments

We would like to acknowledge and sincerely thank our staff who help carried out this research to contribute to the medical research society, including our friends and family, whose encouragement comforted and propelled our work to be done successfully and fantastically. We immensely appreciate both direct and indirect help from all of you.

## Author Contributions

**Conceptualization:** Kawintharat Harirugsakul, Sorawit Wainipitapong, Kavirach Tantiwongse.

**Data curation:** Kawintharat Harirugsakul, Leilani Paitoonpong.

**Formal analysis:** Jeerath Phannajit.

**Investigation:** Kawintharat Harirugsakul, Sorawit Wainipitapong, Leilani Paitoonpong, Kavirach Tantiwongse.

**Methodology:** Kawintharat Harirugsakul, Sorawit Wainipitapong, Jeerath Phannajit, Leilani Paitoonpong, Kavirach Tantiwongse.

**Project administration:** Kavirach Tantiwongse.

**Resources:** Leilani Paitoonpong.

**Supervision:** Sorawit Wainipitapong, Jeerath Phannajit, Leilani Paitoonpong, Kavirach Tantiwongse.

**Validation:** Kawintharat Harirugsakul, Sorawit Wainipitapong.

**Visualization:** Kawintharat Harirugsakul.

**Writing – original draft:** Kawintharat Harirugsakul, Sorawit Wainipitapong.

**Writing – review & editing:** Kawintharat Harirugsakul, Sorawit Wainipitapong, Jeerath Phannajit, Leilani Paitoonpong, Kavirach Tantiwongse.

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
