## [Decision Letter · Decision Letter 0]

20 Jul 2022

PONE-D-22-17958Erectile dysfunction after COVID-19 infection: A follow-up studyPLOS ONE

Dear Dr. Tantiwongse,

Thank you for submitting your manuscript to PLOS ONE. After careful consideration, we feel that it has merit but does not fully meet PLOS ONE’s publication criteria as it currently stands. Therefore, we invite you to submit a revised version of the manuscript that addresses the points raised during the review process.

We look forward to receiving your revised manuscript.

Kind regards,

Taeyun Kim

Academic Editor

PLOS ONE

2. Please specify at which institutions (e.g. hospital or University) the study was conducted

3. PLOS ONE follows the WHO criteria for clinical trails, and during the internal evaluation of your mansucript we did not feel that the study met this criteria. Therefore we would recommend excluding all clinical trial references within the mansucript text to avoid confusion.

5. We note you have included a table to which you do not refer in the text of your manuscript. Please ensure that you refer to Table 3 in your text; if accepted, production will need this reference to link the reader to the Table.

Additional Editor Comments:

* This paper provides very useful information and implicates the possible relationship between ED and COVID-19, although I would like to suggest several points especially regarding the methodology.

* Materials and methods; This section is very important, although lots of researchers would not pay much attention to the methodology the authors utilized. In this regard, I suggest to be more specific with the subheadings as 1) Study design and participants, 2) Data collection (in this part, authors can use sub-subheadings such as 2-1) study design, 2-2) demographic data, 2-3) assessment of erectile function), and 3) Statistical analysis.

* Statistical analysis; Please be specific on the statistical method they used to calculate the significant level. It seems vague to just state: "~~ tests were used regarding variables' nature and distribution.

* Statistical analysis; It seems they used one-way anova test. However, to compare the values within the repeated measurements, RM-ANOVA would be more accurate. Furthermore, the linear mixed-effect model would be more accurate than RM-ANOVA, because this method requires several fastidious assumptions. Therefore to confirm the effect of covid-19 on the ED, which was dichotomized by the score like in Table 2, I suggest authors using the LMEM model. It would be more informative to be visualized.

* Please explain the meaning of round brakets used in Table 1. Also, please indicate the method by which P value was calculated in the footnote of Table 1.

* Please explain the meaning of round and square brakets which were used in Table 2. Also, please indicate the method by which P value was obtained in the footnote of Table 2.

* Please explain the meaning of round and square brakets which were used in Table 4. Also, please indicate the method by which P value was obtained in the footnote of Table 4. In addition, I couldn't find the method by which P value was calculated in the section of Statistical Analysis. Also, please explain the definition of four groups of ED with relevant citations.

* Please discuss that this study did not demonstrate the effect of COVID-19 on the ED in compliance with the Reviewer's comment. And, please discuss other confounding factors that could also impact the ED.

Reviewers' comments:

Reviewer's Responses to Questions

**Comments to the Author**

1. Is the manuscript technically sound, and do the data support the conclusions?

Reviewer #1: Yes

Reviewer #2: Yes

2. Has the statistical analysis been performed appropriately and rigorously? 

Reviewer #1: Yes

Reviewer #2: Yes

3. Have the authors made all data underlying the findings in their manuscript fully available?

Reviewer #1: No

Reviewer #2: Yes

4. Is the manuscript presented in an intelligible fashion and written in standard English?

Reviewer #1: Yes

Reviewer #2: Yes

5. Review Comments to the Author

Reviewer #1: Dear Authors,

Congratulations on your research. I have read the manuscript with keen interest, being this topic one of my main foci of research in the last couple of years, and I am positively impressed by this manuscript.

I have a few minor suggestions.

Line 70: I believe that our study https://pubmed.ncbi.nlm.nih.gov/33234430/ would be a better fit than the reference used here. It's also worth mentioning that sexual health and depression/anxiety, while bidirectionally associated, follow a particular relationship: indeed, being more sexually active can improve psychological health more than the opposite, suggesting that in fact COVID patients with more depressed/anxious mood could also have had experienced negative effects of restriction measures (e.g. lockdowns) for sexual health.

Lines 86-87: missing a '

Reviewer #2: The study appears to be sound, and it is clearly designed and written. The statistics are well done. The article treats an actual problem related to COVID-19, as sexuality and functional sexuality, in this case: ED.

The Title is providing a distorted understanding, giving the impression that ED would be due to COVID-19. The authors should clarify the title section to avoid this confusion.

As this study determines, the age and mental status, the major depression due to health problems clearly influence the erectile functioning. There is no evidence that in this population there would be a direct relationship between the virus and the ED. Neither is there a reference group, formed from subjects that would have been severely ill, and long-time hospitalized, for other reasons than COVID-19, but with same demographic characteristics.

In this study, the only illness discussed is COVID-19. Evidently, the results would come as for patients recovering from COVID-19, not necessary having any relationship between the SARS-CoV-2 viral infection and ED.

Additionally, as it results from this study, the morning erection was normal both at baseline and after 3 months follow-up (p=0,13).

To admit for publication the title should be changed, as the authors did not study different groups of severe illnesses and prolonged hospitalization, which should have been studied in order to be able to say that ED is indeed after COVID-19 infection and not just related to age and major depression as an effect of a difficult illness.

The first sentence of the second paragraph from Discussion is not clear. The authors should indicate precisely what they are referring to with “it”. Please replace it with the exact meaning.

The topic of the study is of great importance. In the Discussion part, the authors clarified well the ambiguous relation between ED and COVID-19. This is not yet seen in the title. This study brings a great potential for further studies on the topic.

6. PLOS authors have the option to publish the peer review history of their article (what does this mean?). If published, this will include your full peer review and any attached files.

Reviewer #1: No

Reviewer #2: **Yes: **Dr Corina Rosemarie Iire

---

## [Author Response · Author response to Decision Letter 0]

18 Aug 2022

Dear Editor and Reviewers,

Thank you for your letter and the opportunity to revise our manuscript. All suggestions offered by the editor and reviewers have been immensely helpful, and we appreciate the insightful comments on every aspect of the paper. We are happy to inform you that all the suggestions are complied with and detailed below. We hope that the revised manuscript will better suit for publication.

Regards,

On behalf of all authors,

Kavirach Tantiwongse

August 19, 2022

Response: Our revised manuscript was edited to comply with PLOS ONE’s style requirements.

2. Please specify at which institutions (e.g. hospital or University) the study was conducted. 

Response: Thank you for your comment. We have added the name of our study location in the method section. (Page 5 Line 115-116)

3. PLOS ONE follows the WHO criteria for clinical trials, and during the internal evaluation of your mansucript we did not feel that the study met this criteria. Therefore we would recommend excluding all clinical trial references within the mansucript text to avoid confusion.

Response: Thank you for your suggestion. Clinical trial references have been removed.

4. In your Data Availability statement, you have not specified where the minimal data set underlying the results described in your manuscript can be found. PLOS defines a study's minimal data set as the underlying data used to reach the conclusions drawn in the manuscript and any additional data required to replicate the reported study findings in their entirety. All PLOS journals require that the minimal data set be made fully available. 

Response: Thank you for this point. Since our dataset consists of sensitive information, identifiable data of all participants have been removed. We have deposited our data set in the repository site and mentioned about this in the manuscript. (Page 23 Line 476-478)

5. We note you have included a table to which you do not refer in the text of your manuscript. Please ensure that you refer to Table 3 in your text; if accepted, production will need this reference to link the reader to the Table.

Response: Thank you. The text referring to Table 3 has been added. (Page 11 Line 234)

Additional Editor Comments:

* Materials and methods; This section is very important, although lots of researchers would not pay much attention to the methodology the authors utilized. In this regard, I suggest to be more specific with the subheadings as 1) Study design and participants, 2) Data collection (in this part, authors can use sub-subheadings such as 2-1) study design, 2-2) demographic data, 2-3) assessment of erectile function), and 3) Statistical analysis.

Response: Thank you very much for these positive comments and suggestion. We agree with the editor that the methods section is very important and we try our best to add more important information. We have now modified the subheadings and sub-subheadings within the materials and methods regarding your suggestions in our revised manuscript. (Page 6 Line 127 - 

Page 9 Line 191)

* Statistical analysis; Please be specific on the statistical method they used to calculate the significant level. It seems vague to just state: "~~ tests were used regarding variables' nature and distribution.

Response: Thank you. We have specified the statistical methods used in our study and they have been rearranged appropriately in the statistical analysis sub-subheading. (Page 8 Line 167 – 

Page 9 Line 191)

* Statistical analysis; It seems they used one-way anova test. However, to compare the values within the repeated measurements, RM-ANOVA would be more accurate. Furthermore, the linear mixed-effect model would be more accurate than RM-ANOVA, because this method requires several fastidious assumptions. Therefore to confirm the effect of covid-19 on the ED, which was dichotomized by the score like in Table 2, I suggest authors using the LMEM model. It would be more informative to be visualized.

Response: Thank you for your suggestion. In this revised version, we use the linear mixed effect model for continuous repeated measurements (including IIEF-5 score, GAD and PHQ) and generalized linear mixed effect model for binary data (ED status, morning erection, depression, and anxiety). The model is comprised of visit (baseline vs. three-month) as a co-variate and patient ID as fixed intercept in order to demonstrate the temporal effects on each variable after recovery from COVID-19. The detailed of models are described in ‘Statistical analysis’ section (Page 8 Line 167 - Page 9 Line 191) and results are presented in table 2.

* Please explain the meaning of round brakets used in Table 1. Also, please indicate the method by which P value was calculated in the footnote of Table 1.

* Please explain the meaning of round and square brakets which were used in Table 2. Also, please indicate the

method by which P value was obtained in the footnote of Table 2.

* Please explain the meaning of round and square brakets which were used in Table 4. Also, please indicate the method by which P value was obtained in the footnote of Table 4. In addition, I couldn't find the method by which P-value was calculated in the section of Statistical Analysis. 

Response: Apologize for the initially confusing tables, we have edited all suggested and added footnotes on each table explaining all abbreviations, data presentation, and statistical analysis methods. (please see table 1, 2 and 4)

* Also, please explain the definition of four groups of ED with relevant citations.

Response: Our study categorized patients into four groups based on the onset of ED relating with COVID-19. However, up to our knowledge, there is still no standard definition to categorize the ED related to COVID-19. We have discussed more about this in our revised manuscript. (Page 12 Line 252-253)

* Please discuss that this study did not demonstrate the effect of COVID-19 on the ED in compliance with the Reviewer's comment. And, please discuss other confounding factors that could also impact the ED.

Response: Thank you for the crucial point. According to our methodology, we are not able to conclude the direct effect of COVID-19 and ED as the reviewer’s comment. However, our study could imply that ED improvement could be found after COVID-19 recovery and some associated factors are worth to be mentioned. Because of the complexity of ED etiologies, several confounding factors might impact the ED and we have discussed about this more in the discussion section as your recommendation. (Page 16 Line 325-327)

Reviewer's Responses to Questions

Comments to the Author

1. Is the manuscript technically sound, and do the data support the conclusions?

Reviewer #1: Yes

Reviewer #2: Yes

2. Has the statistical analysis been performed appropriately and rigorously? 

Reviewer #1: Yes

Reviewer #2: Yes

3. Have the authors made all data underlying the findings in their manuscript fully available?

Reviewer #1: No

Response: Now, we have uploaded our dataset in the repository and provided the available link in the data availability section of our revised manuscript. (Page 23 Line 476-478)

Reviewer #2: Yes

4. Is the manuscript presented in an intelligible fashion and written in standard English?

Reviewer #1: Yes

Reviewer #2: Yes

Reviewer #1: Dear Authors,

Congratulations on your research. I have read the manuscript with keen interest, being this topic one of my main foci of research in the last couple of years, and I am positively impressed by this manuscript.

I have a few minor suggestions.

Line 70: I believe that our study https://pubmed.ncbi.nlm.nih.gov/33234430/ would be a better fit than the reference used here. It's also worth mentioning that sexual health and depression/anxiety, while bidirectionally associated, follow a particular relationship: indeed, being more sexually active can improve psychological health more than the opposite, suggesting that in fact COVID patients with more depressed/anxious mood could also have had experienced negative effects of restriction measures (e.g. lockdowns) for sexual health.

Response: Thank you. We appreciate your helpful suggestion. We found your study suit our finding and have now added this in the introduction part and references. (Page 3 Line 70). Moreover, we also added information about bidirectional effects in sexual health and psychological health in the discussion part. (Page 14 Line 299 - Page 15 Line 301)

Lines 86-87: missing a '

Response: Thank you. This mistake has been corrected. (Page 4 Line 87)

Reviewer #2: The study appears to be sound, and it is clearly designed and written. The statistics are well done. The article treats an actual problem related to COVID-19, as sexuality and functional sexuality, in this case: ED.

The Title is providing a distorted understanding, giving the impression that ED would be due to COVID-19. The authors should clarify the title section to avoid this confusion.

Response: Thank you for this comment. We have renamed our title to ‘Erectile Dysfunction after COVID-19 Recovery: A follow-up study’ according to your suggestion. (Page 1 Line 1)

As this study determines, the age and mental status, the major depression due to health problems clearly influence the erectile functioning. There is no evidence that in this population there would be a direct relationship between the virus and the ED. Neither is there a reference group, formed from subjects that would have been severely ill, and long-time hospitalized, for other reasons than COVID-19, but with same demographic characteristics. In this study, the only illness discussed is COVID-19. Evidently, the results would come as for patients recovering from COVID-19, not necessary having any relationship between the SARS-CoV-2 viral infection and ED. Additionally, as it results from this study, the morning erection was normal both at baseline and after 3 months follow-up (p=0,13).

Response: Thank you for this point. We have mentioned the limitation of our study design that cannot conclude the direct effect of COVID-19 and ED. (Page 16 Line 325-327)

To admit for publication the title should be changed, as the authors did not study different groups of severe illnesses and prolonged hospitalization, which should have been studied in order to be able to say that ED is indeed after COVID-19 infection and not just related to age and major depression as an effect of a difficult illness.

Response: Thank you. We greatly appreciate your comment helping us to improve our manuscript. Our title has been changed to comply with our study design and reduce confusion regarding your suggestion.

The first sentence of the second paragraph from Discussion is not clear. The authors should indicate precisely what they are referring to with “it”. Please replace it with the exact meaning.

Response: Thank you for this point. We have replaced ‘it’ with the referred word ‘long COVID’. (Page 13 Line 276)

The topic of the study is of great importance. In the Discussion part, the authors clarified well the ambiguous relation between ED and COVID-19. This is not yet seen in the title. This study brings a great potential for further studies on the topic.

Response: Thank you for your helpful comments and hope that our revised manuscript and renamed title will suit for the publication.

---

## [Decision Letter · Decision Letter 1]

3 Oct 2022

PONE-D-22-17958R1Erectile Dysfunction after COVID-19 Recovery: A Follow-Up StudyPLOS ONE

Dear Dr. Tantiwongse,

Thank you for submitting your manuscript to PLOS ONE. After careful consideration, we feel that it has merit but does not fully meet PLOS ONE’s publication criteria as it currently stands. Therefore, we invite you to submit a revised version of the manuscript that addresses the points raised during the review process.

We look forward to receiving your revised manuscript.

Kind regards,

Taeyun Kim

Academic Editor

PLOS ONE

Journal Requirements:

Additional Editor Comments:

* I have a few minor comments regarding the Abstract.

* Abstract Method; please briefly introduce the main methods that was used in the current study: mixed-effect model and logistic regression model.

* Abstract Results; This section should be in line with the methods section. That is, when method A is introduced in Method section, the results from this method also should be described in Results section.

* Abstract; If the word count exceeds, introduction section could be simplified, just stating "The present study aimed to ~".

Reviewers' comments:

Reviewer's Responses to Questions

**Comments to the Author**

1. If the authors have adequately addressed your comments raised in a previous round of review and you feel that this manuscript is now acceptable for publication, you may indicate that here to bypass the “Comments to the Author” section, enter your conflict of interest statement in the “Confidential to Editor” section, and submit your "Accept" recommendation.

Reviewer #1: All comments have been addressed

Reviewer #2: All comments have been addressed

2. Is the manuscript technically sound, and do the data support the conclusions?

Reviewer #1: Yes

Reviewer #2: Yes

3. Has the statistical analysis been performed appropriately and rigorously? 

Reviewer #1: Yes

Reviewer #2: Yes

4. Have the authors made all data underlying the findings in their manuscript fully available?

Reviewer #1: Yes

Reviewer #2: Yes

5. Is the manuscript presented in an intelligible fashion and written in standard English?

Reviewer #1: Yes

Reviewer #2: Yes

6. Review Comments to the Author

Reviewer #1: I wish to thank the Authors for addressing all comments I raised during the first iteration of peer review. I have no further remarks.

Reviewer #2: Dear Authors,

I appreciate the corrections and additions were done to the manuscript, as improving the description of your statistical analysis, tables and all together the “Materials and Methods” section. Also the title is a better one in cleaning the previous confusion. I thank you for the rigorous work on your study, with a very important topic. That is a good study.

7. PLOS authors have the option to publish the peer review history of their article (what does this mean?). If published, this will include your full peer review and any attached files.

Reviewer #1: No

Reviewer #2: No

---

## [Author Response · Author response to Decision Letter 1]

4 Oct 2022

Response to Editor and Reviewers

Dear Editor and Reviewers,

We appreciate all your kind comments and suggestions. Hereby, please find our point-by-point responses listed below.

Regards,

On behalf of all authors,

Kavirach Tantiwongse

October 4, 2022

Response: All references have been reviewed, and none of them has been retracted. 

Additional Editor Comments:

* Abstract Method; please briefly introduce the main methods that was used in the current study: mixed-effect model and logistic regression model.

Response: Thank you for this important point. We have now mentioned both generalized linear mixed-effect model (GLMM) and logistic regression, the main methods of our study, in the abstract, according to your suggestion. (Page 2 Line 35-38)

* Abstract Results; This section should be in line with the methods section. That is, when method A is introduced in the Method section, the results from this method also should be described in Results section.

Response: Thank you for your helpful advice. Additional results from GLMM have been added. (Page 2 Line 39-44)

* Abstract; If the word count exceeds, introduction section could be simplified, just stating "The present study aimed to ~".

Response: Regarding all changes that have been made, the total word count is 287 for the abstract.

---

## [Editor Report · Decision Letter 2]

7 Oct 2022

Erectile Dysfunction after COVID-19 Recovery: A Follow-Up Study

PONE-D-22-17958R2

Dear Dr. Tantiwongse,

We’re pleased to inform you that your manuscript has been judged scientifically suitable for publication and will be formally accepted for publication once it meets all outstanding technical requirements.

Kind regards,

Taeyun Kim

Academic Editor

PLOS ONE
---

## [Editor Report · Acceptance letter]

12 Oct 2022

PONE-D-22-17958R2 

Erectile Dysfunction after COVID-19 Recovery: A Follow-Up Study 

Dear Dr. Tantiwongse:

I'm pleased to inform you that your manuscript has been deemed suitable for publication in PLOS ONE. Congratulations! Your manuscript is now with our production department. 

Kind regards, 

on behalf of

Dr. Taeyun Kim 

Academic Editor

PLOS ONE